# Nutritional and Functional New Perspectives and Potential Health Benefits of Quinoa and Chia Seeds

**DOI:** 10.3390/antiox12071413

**Published:** 2023-07-12

**Authors:** Aparna Agarwal, Abhishek Dutt Tripathi, Tarika Kumar, Kanti Prakash Sharma, Sanjay Kumar Singh Patel

**Affiliations:** 1Department of Food & Nutrition and Food Technology, Lady Irwin College, Sikandra Road, New Delhi 110001, India; aparna.gupta@lic.du.ac.in; 2Department of Food Technology, Bhaskaracharya College of Applied Sciences, Sector-2, Dwarka, New Delhi 110075, India; 3Department of Dairy Science and Food Technology, Institute of Agricultural Sciences, Banaras Hindu University, Varanasi 221005, India; 4Department of Environmental Studies, The Maharaja Sayajirao University of Baroda, Vadodara 390002, India; 5Department of Nutrition Biology, Central University of Haryana, Mahendergarh 123031, India; 6Department of Chemical Engineering, Konkuk University, Seoul 05029, Republic of Korea

**Keywords:** antioxidants, bioactive components, chia seed, quinoa, nutritional properties, health benefits, medicinal values

## Abstract

Quinoa (*Chenopodium quinoa* Willd) and chia (*Salvia hispanica*) are essential traditional crops with excellent nutritional properties. Quinoa is known for its high and good quality protein content and nine essential amino acids vital for an individual’s development and growth, whereas chia seeds contain high dietary fiber content, calories, lipids, minerals (calcium, magnesium, iron, phosphorus, and zinc), and vitamins (A and B complex). Chia seeds are also known for their presence of a high amount of omega-3 fatty acids. Both quinoa and chia seeds are gluten-free and provide medicinal properties due to bioactive compounds, which help combat various chronic diseases such as diabetes, obesity, cardiovascular diseases, and metabolic diseases such as cancer. Quinoa seeds possess phenolic compounds, particularly kaempferol, which can help prevent cancer. Many food products can be developed by fortifying quinoa and chia seeds in different concentrations to enhance their nutritional profile, such as extruded snacks, meat products, etc. Furthermore, it highlights the value-added products that can be developed by including quinoa and chia seeds, alone and in combination. This review focused on the recent development in quinoa and chia seeds nutritional, bioactive properties, and processing for potential human health and therapeutic applications.

## 1. Introduction

Due to urbanization, the lifestyle of people is changing rapidly, which inclines them towards healthy living. Nowadays, people are more conscious about their eating patterns, such as what they eat and whether they eat healthy foods. So, in the era of healthy eating, foods with excellent nutritional benefits play a vital role as they provide nourishment to the human body for growth and development and, in addition to this, help in preventing various chronic diseases [1,2,3,4]. Today, non-communicable diseases such as cancer, cardiovascular diseases, and diabetes are increasing faster. Globally, two out of three deaths are caused yearly due to non-communicable diseases. Also, traditional crop usage has declined with time, although they have outstanding medicinal properties [5,6,7]. Using processed and ready-to-eat foods such as maize extrudates by incorporating chia and quinoa seeds can help utilize novel ingredients for good organoleptic properties and health benefits [8,9,10]. Therefore, quinoa and chia are two such traditional and ancient crops that are considered to be superfoods as they possess various nutritional benefits such as antioxidant activity, antidiabetic, anticancer, antihypertensive, antimicrobial, antithrombotic peptides, and functional food additives due to the bioactive compounds present in them [11,12,13,14,15,16]. With the recent pandemics and the continuous onslaught of outbreaks, the focus is shifted toward coordinating global efforts on high-risk public health issues through various therapeutic strategies, including natural products [17,18,19,20,21,22]. Recent reviews on numerous aspects of quinoa and chia seeds—properties and potential therapeutic applications—can be seen [23,24,25,26,27,28]. This review elaborates on the properties and health benefits of quinoa and chia seeds. Furthermore, this study also focuses on the various nutritional and bioactive properties of chia and quinoa seeds and the value addition of both in the food products for a sustainable diet.

## 2. Quinoa and Its Characteristics

Quinoa is a plant scientifically known as *Chenopodium quinoa* Willd and belongs to the Chenopodiaceae family [29]. It is known as “mother grain” by the Incas and is considered a sacred plant that contributes various medicinal properties. Quinoa is cultivated around the globe with 250 varieties and is usually 1 to 3 m in height, and native to the Andean regions of South America, such as Peru, Bolivia, Chile, Argentina, Ecuador, and Colombia [30,31,32]. In addition, it is also grown in Colorado, Washington, Oregon, England, and Scandinavia. Quinoa is a pseudocereal consisting of seeds that can be pulverized in flour and used as a cereal crop. It is gaining a lot of attention as it is resistant to different temperatures and climatic conditions, such as it can be grown and survive in areas of low rainfall, temperate regions, marginal lands, high temperature, and poor sandy, alkaline soils and yet maintain its high nutritional value [33,34,35]. Due to its high micronutrient intake, resistance to extreme climatic conditions, dietary benefits, and fewer inputs for production, such as chemical fertilizers, quinoa is considered a “future smart food” by the Food and Agriculture Organization of the United Nations (FAO) [36]. It consists of leaves and seeds as edible parts, but seeds are gaining more importance due to their nutritional benefits [29]. Quinoa seeds show a high composition of proteins, carbohydrates, and lipids with excellent balance [34,37]. It supplies all the amino acids as FAO/WHO recommends for an adult. Due to the high nutritional value of quinoa, the FAO declared 2013 as the “Year of Quinoa” [38].

Quinoa seeds are round and flat in appearance. They have a diameter ranging from 1.5 to 4 mm and consist of various sizes and colors. The color of the seeds is usually black, yellow, white, red, purple, or violet, depending upon the variety. Although the seeds have different sizes and colors, they fall under single species [39,40]. Furthermore, quinoa is also gaining importance in India under a project named “Anantha” by cultivating quinoa in Hyderabad and Ananthapuramu in Andhra Pradesh [41].

### 2.1. Nutrients in Quinoa Seeds

Quinoa is known as a complete food due to its high nutritional value, i.e., proteins, carbohydrates, lipids, vitamins, and minerals [26,42]. Table 1 provides the composition of the various nutrients present in uncooked quinoa. The protein content in quinoa seeds varies from 13.8 to 16.5%, an average of 15%. Usually, albumins account for 35%, and globulins account for 37% of total proteins; these are the major proteins present in quinoa, although it also consists of a low amount of prolamins [29]. Therefore, the content of amino acids present in the quinoa seeds plays a vital role in determining the nutritional characteristic of protein [26]. Quinoa is known for the presence of all nine essential amino acids [i.e., phenylalanine, methionine, histidine (key role in childhood), isoleucine, valine, leucine, lysine, threonine, tryptophan], which are essential for humans in their growth and development and provide a similar protein efficiency ratio to that of milk casein. Also, due to the absence of gliadins and gliadin-related protein fractions (usually present in rye, barley, and malt), it is classified as gluten-free and considered suitable for celiac disease patients [43].

Starch is the primary carbohydrate in quinoa, about 58.1 to 64.2%, out of which 11% is amylose content [26,39]. It is present in various sizes, such as large (>15 µm), medium (5–15 µm), and small (<5 µm). Also, the diameter of starch granules is tiny compared to other cereals such as wheat, rice, and amaranth, which makes it helpful in providing enhanced processing ability and better quality of food products, e.g., noodles. Quinoa starch is also used in making edible and recyclable films. These films have good mechanical and barrier properties and low solubility, making them ideal for food packaging [44]. The dietary fiber content in quinoa ranges from 7.0 to 9.7% (dry matter), which is very near to the amount present in cereals, in which soluble dietary fiber content ranges from 1.3 to 6.1% (dry weight). In addition, simple sugars, including maltose, D-galactose, and D-ribose, are also present at around 3% in quinoa with less fructose and glucose levels [45].

**Table 1 antioxidants-12-01413-t001:** Composition of nutrients in quinoa and chia seeds [24,25,29,46].

Nutritional Components	Unit	Availability (Per 100 g)	
		Quinoa	Chia
Water	g	13.3	4.43
Energy	kcal	368	486
Protein	g	14.1	24.2
Total Fat	g	6.10	40.2
Saturated Fatty Acids	g	0.70	5.00
Monounsaturated Fatty Acids	g	1.60	2.96
Polyunsaturated Fatty Acids	g	3.30	22.8
Dietary Fiber	g	7.00	34.4
Carbohydrates	g	64.2	42.1
Ash	g	2.40	4.8
Essential amino acids			
Phenylalanine	g	0.59	1.02
Methionine	g	0.31	0.59
Histidine	g	0.41	0.53
Isoleucine	g	0.50	0.80
Valine	g	0.59	0.95
Leucine	g	0.84	1.37
Lysine	g	0.77	0.97
Threonine	g	0.42	0.71
Tryptophan	g	0.17	0.44
Arginine	g	0.03	2.14
Non-essential amino acids			
Cystine	g	0.19	0.41
Tyrosine	g	0.52	0.56
Aspartic acid	g	1.45	1.69
Glutamic acid	g	12.8	3.50
Alanine	g	0.64	1.04
Glycine	g	0.30	0.94
Proline	g	0.86	0.78
Serine	g	0.47	1.05
Vitamins			
Thiamin (B_1_)	mg	0.36	0.60
Riboflavin (B_2_)	mg	0.32	0.20
Niacin (B_3_)	mg	1.52	8.80
Pantothenic acid (B_5_)	mg	5.60	0.94
Pyridoxine (B_6_)	mg	0.49	0.10
Folic acid (B_9_)	mg	6.50	49.0
Vitamin B_12_	mg	0.23	0.02
Vitamin A	mg	14.0	54.0
ꞵ-carotene	µg	8.00	10.2
α-carotene	µg	0.00	0.00
ꞵ-cryptoxanthin	µg	1.00	0.00
Lutein + Zeaxanthin	µg	163	68.0
α-tocopherol	mg	2.44	3.05
ꞵ-tocopherol	mg	0.08	0.08
γ-tocopherol	mg	4.55	5.53
δ-tocopherol	mg	0.35	1.57
Vitamin K	mg	0.00	0.00
Vitamin D (D_2_ + D_3_)	mg	15.1	0.00
Vitamin C	mg	10.2	1.60
Vitamin E	mg	4.15	0.50
Minerals			
Calcium	mg	47.0	631
Iron	mg	4.6	7.7
Phosphorus	mg	457	860
Sodium	mg	5.0	16.0
Potassium	mg	563	407
Magnesium	mg	197	335
Copper	mg	0.6	0.9
Zinc	mg	3.1	4.6
Manganese	mg	2.0	2.7
Selenium	µg	8.5	0.04

Due to the rich quality and quantity of lipid fraction in quinoa, they are referred to as an alternative oilseed crop. Oil content in quinoa varies from 2.0 to 9.5%, with a rich source of essential fatty acids such as linoleic and α-linolenic acids. In addition, quinoa is also a rich source of α-tocopherol and γ-tocopherol, which act as antioxidants to prevent lipid oxidation [43,47]. Oil content in quinoa seeds consists of approximately 89.4% of unsaturated fatty acids, out of which 54.2 to 58.3% are polyunsaturated fatty acids (PUFAs) [48]. In addition, linoleic acid is abundantly present in quinoa seeds, providing various health benefits such as preventing cardiovascular diseases [47].

Quinoa is a rich source of riboflavin (Vitamin B2), pyridoxine (Vitamin B6), and folic acid (Vitamin B9). Folic acid and pyridoxine in 100 g of quinoa have been proven to meet normal adult dietary needs. Also, in 100 g of quinoa, riboflavin met 80% of children’s and 40% of adults’ nutritional requirements [29]. Quinoa is also a rich source of minerals such as magnesium, copper, calcium, phosphorus, potassium, and iron. These metals are vital for various biological activities and for improving human health [26,49]. It has higher magnesium content, i.e., 0.26%, than wheat and corn, i.e., 0.16 and 0.14%, respectively [39]. Minerals are vital in providing benefits to the human body, such as magnesium, which helps control the blood sugar level and reduces Type-2 diabetes in patients. Also, potassium is vital for a healthy heart and lowering blood pressure [50]. The mineral content of quinoa seeds is more significant than oat and barley in terms of calcium, potassium, and magnesium. Magnesium, manganese, copper, and iron in 100 g of quinoa seeds are proven to help fulfill the daily needs of infants and adults. However, potassium in 100 g of quinoa seeds can satisfy only 18% of infant and 22% of adult dietary needs [47].

### 2.2. Bioactive Compounds in Quinoa and Its Biological Properties

Quinoa is well known for its high nutritional content. In addition, it also possesses medicinal health benefits against different chronic diseases such as diabetes, obesity, anemia, and celiac disease [5,26]. These health benefits are due to certain phytochemicals that make quinoa superior to other grains in providing human health and wellness (Figure 1). Various bioactive compounds in quinoa provide medicinal properties, such as phytosterols, saponins, phenolic compounds, phytoecdysteroids, polysaccharides, and betalains [24,51]. Neumerous varieties of quinoa (white, red, and black) have been found to be rich in phenolic compounds, especially flavonoids, which benefit humans. Studies have proven that quinoa has not been seen to have toxicity [52]. Artificial intelligence or proteomics mining strategies can be potentially employed to identify novel protein perspective immuno-nutritional bioactivities [53]. In addition, the microbial pure or co-culture fermentation of quinoa seed’s therapeutic properties, such as antioxidant activity and antidiabetic influence, can potentially alter [54]. High concentrations of proanthocyanidins have been found in quinoa; these are oligomeric flavonoids with antioxidant, antidiabetic, anti-inflammatory, and anticancer properties [24,55,56]. The seed quality and composition of quinoa are highly varied under hyper-arid desert environments, and different locations of Canada’s growth conditions [57,58].

#### 2.2.1. Phytosterols

Phytosterols are the lipophilic compounds in the quinoa seeds that help reduce the cholesterol level in the body, as illustrated in Figure 2. The structure makes phytosterols identical to cholesterol [59]. The primary mechanism involves the phytosterols competing for the cholesterol’s intestinal absorption. After the breakdown of the phytosterols in the small intestine, they replace the cholesterol in the micelles and thus reduce the production of low-density lipoprotein (causes fatty deposits in the arteries) in the intestine and liver to ultimately reduce the serum cholesterol in the body. Phytosterols of 118 mg are present per 100 g of quinoa seeds [26,29]. In addition to the cholesterol-lowering effect, phytosterols also play a vital role in providing other biological influences, such as anti-carcinogenic activity, anti-inflammatory properties, and antioxidative properties. The significant phytosterols in quinoa seeds are β-sitosterol, campesterol, and stigmasterol, with concentrations of 63.7, 15.6, and 3.2 mg/100 g, respectively [47].

#### 2.2.2. Saponins

Saponins are the secondary metabolites of plants with biological characteristics such as protection against pathogens, pests, and insects. These are considered anti-nutritional factors mainly found in the seed coat of quinoa (about 86% of total saponins present in seeds). There are 30 varieties of saponins found in the various parts of the quinoa plant [60]. Saponins possess various medicinal benefits, such as antioxidant activity, antimicrobial agents, anti-inflammatory properties, and cytotoxic effects [61]. The anti-inflammatory effect of quinoa is majorly due to the presence of monodesmosidic saponin present in its grains. It helps prevent oxidative damage and increased inflammation which can otherwise lead to chronic diseases such as cardiovascular diseases, neurological diseases, and even cancer. Saponins provide sweet as well as bitter tastes depending upon their content present. The sweet taste of saponins is usually associated with the saponins content of 20–40 mg/g of dry weight and bitter taste (present in the outer seed coat) with >470 mg/g of dry weight. Saponins can be removed by processing methods such as washing, dehulling, and abrasion [26].

#### 2.2.3. Phenolics

Phenolic compounds are considered secondary plant metabolites that contain hydroxyl groups attached to at least one aromatic hydrocarbon ring. Although many phenolic compounds exist in a plant, such as phenolic acids, flavonoids, phenols, quinines, coumarins, lignans, phenylpropanoids, and xanthones [62], but flavonol glycosides mainly consist of quercetin and kaempferol, most abundantly present in quinoa. Polyphenol content in quinoa seeds varies between 0.46 and 1.84 mg/g of dry weight. Other phenolic acids, such as ferulic, caffeic, and *p*-coumaric acids with 251.5, 6.31, and 1.1 µg/g of dry weight, respectively, are also present in quinoa seeds [24]. Quinoa seeds possess greater antioxidant capacity due to their presence of phenolic compounds. The antioxidant helps in scavenging free radicals and preventing oxidative stress.

Phenolic antioxidants are in free and bound forms attached to the cell wall [63]. Kaempferol in quinoa seeds helps to prevent cancer, as described in Figure 3. Aerobic metabolism or environmental factors lead to byproducts, including reactive oxygen species (ROS), which possess carcinogenic properties by inducing cancer cell growth in the human body [64]. The superoxide anion, which is a member of ROS, is converted into hydrogen peroxide (H_2_O_2_) with the action of superoxide dismutase. Further, H_2_O_2_ is converted into hydroxy radical by reacting with reduced metals. In addition, when it reacts with nitric oxide, H_2_O_2_ also produces peroxynitrite, and all these formed products induce DNA damage and lead to carcinogenesis. Kaempferol has the potential to possess antioxidant as well as anti-inflammatory action to maintain cellular redox homeostasis by scavenging the superoxide anion, hydroxyl radical, and peroxynitrite, which can cause cancer in the human body [64]. Furthermore, during the ROS production leading to oxidative stress and cancer, Nrf2 (a leucine zipper transcription factor that protects the cells from oxidative stress) detached from Keap 1 (protein) and moved towards the nucleus to activate cytoprotective genes to inhibit the oxidative stress-induced cancer. However, under basal state, Keap 1 maintained the functioning of Nrf2 in the cytoplasm and dissociated it during the ubiquitin-proteasome pathway [65]. Therefore, kaempferol inhibits cancer generation by up-regulating the Nrf2 and Keap 1 cell defense pathways [64].

#### 2.2.4. Phytoecdysteroids

Phytoecdysteroids are polyhydroxylated steroids, which are also one of the bioactive compounds present in the highest amount in quinoa seeds, i.e., 138 to 570 µg/g among all other edible crops. These are used to provide defense action in plants from insects. Also, these have other biological properties such as wound healing, antioxidant, anti-diabetic, immunomodulatory, neuroprotective, and anti-depressive activities [62]. There are 13 phytoecdysteroids in quinoa seeds, of which 20-hydroxyecdysone (20HE) is abundantly present (around 62% to 90%). 20HE possesses biological activities vital in treating or preventing metabolic syndrome and post-menopausal disorders [48].

#### 2.2.5. Polysaccharides

Polysaccharides play a vital role as it acts as an antioxidant that scavenges free radicals and lipid oxidation inhibitory agents and provides immunoregulatory functions. It also offers cytotoxic properties against human liver and breast cancer [24].

#### 2.2.6. Betalains

Betalains are nitrogen-containing plant pigments that are considered under the family of phytochemicals of quinoa and possess bioactive properties [66]. These pigments provide yellow, black, and red color to quinoa seeds and the vegetative parts of the plant. Among the betalains pigments, betanin and isobetanin are most abundantly present in quinoa seeds. These pigments are also known for their medicinal health properties, such as antioxidant and anti-inflammatory properties. A study using the DPPH (1, 1-diphenyl-2-picrylhydrazyl) method also revealed that betalains pigments sometimes have antioxidant capacity more potent than polyphenols [26,48]. Betalains pigments are chemically divided into two forms, i.e., red-violet betacyanins and yellow-orange betaxanthins, which combine and make the orange and red shades. The incorporation of these pigments in food products and pharmaceuticals is now officially certified by the European Union (additive E-162) and US FDA. So, quinoa can also extract betalains, providing novel health benefits [24]. Overall, the details of quinoa seeds bioactive compounds are presented in Table 2.

## 3. Chia Seeds and Its Characteristics

Chia is an herbaceous plant that is scientifically referred to as *Salvia hispanica*, belonging to the family of Lamiaceae. Chia is a Spanish word for “chian” or “chien”, which means oily [25,69]. And its scientific name, *S. hispanica*, was given by Carl Linnaeus (1707–1778). This plant grows in an area covering from western Mexico to northern Guatemala. It is 1 m in height, and its leaves are 4–8 cm long and 3–6 cm wide. The optimum conditions for its cultivation include a warm climate, heavy rainfall, and a temperature range from 15 to 30 °C. Chia seeds color varies from red and white to black, belonging to the regions of Lake Patzcuaro, Michoacan, and Mexico, and are widely used for their health benefits [70]. These are planted in greenhouses in various countries, such as Europe, where it is impossible to grow them in fields due to unfavorable climatic conditions. Also, unfavorable environmental factors such as temperature, light, increased atmospheric CO_2_ level, soil, and nutrition alter seed oil’s protein, quality, and quantity, ultimately hindering the composition of chia seeds. Chia seed yields and oil composition are highly influenced using bio-growth stimulants [71]. The seeds can be utilized in whole, grounded, and milled forms. Also, chia seeds have a mucilaginous texture. This happens when the seeds are in contact with an aqueous medium; it forms a gelatinous mucilage layer around the seeds [72]. The chia seeds in food products, especially meat products, can be used as chia flour, oil, and mucilage (Figure 4). Chia flour is processed by grounding the chia seeds, which are further processed with water and oil to form o/w emulsion, whereas when processed with water, oil, and gelling-agent forms o/w emulsion gel. The oil extraction of chia seeds can be converted into defatted chia flour and chia oil [27,73]. However, bioactive compounds from the oils of chia and quinoa seeds can be extracted using different solvents as shown in Table 3. The chia oil can be processed with alginate through atomization to form microparticles. Furthermore, chia oil can be combined with water, flour, and a gelling agent to form an emulsion gel. Additionally, chia seeds, when soaked in water, form chia mucilage, which, through freeze drying, can be used as powder and, if mixed with oil and gelling agent, can be used as an emulsion gel [73]. The recovery of mucilage is highly influenced by the combined treatment of ultrasound and heat [74]. Chia seeds have been proven to have health benefits such as improved blood lipid profile, hypotensive, hypoglycaemic, antioxidant, antimicrobial, and immunostimulatory effects [25,69,75,76]. The bioactive multi-functional analysis of chia seeds by molecular and in silico approaches, such as peptidomes and molecular dynamics simulations, can be potentially used for broad biotechnological applications [77]. In addition, in silico peptide variants of chia can be predicted with desirable properties such as antioxidant, anti-biofilm, and antimicrobial that can be functionally validated [78,79]. The details of chia seeds bioactive compounds are presented in Table 2.

### 3.1. Nutrients in Chia Seeds

Chia seeds are considered under-utilized pseudocereal even though they contain various nutrients such as protein, dietary fiber, lipids, vitamins, and minerals [80]. However, it is also the richest source of omega-3 fatty acids [81]. The detailed composition of nutrients present in chia seeds is shown in Table 1. Chia seeds contain 15 to 23% protein content, higher than the other cereals such as wheat, oats, barley, corn, rice, quinoa, and amaranth. Also, due to the absence of gluten in chia seeds, it is recommended for celiac disease patients [82]. In addition, the seeds possess 10 exogenous amino acids, of which arginine, leucine, valine, lysine, and phenylalanine are present in more significant amounts. Also, chia seeds are a rich source of endogenous amino acids, including serine, glycine, alanine, glutamic, and aspartic acids [25]. The seeds contain 30 to 34 g of dietary fiber per 100 g, of which 85–93% consists of insoluble dietary fiber and 7–15% of soluble dietary fiber [25]. Therefore, it fulfills the daily requirements of the adult dietary fiber intake, i.e., 25–35 g/day. The fiber content of chia seeds is also higher than quinoa (7.0 g/100 g), flaxseed (27.3 g/100 g), and amaranth (6.7 g/100 g) [82]. The supply of lipids in food is essential for an organism as it helps to provide energy and maintains physiological activities. Chia seeds contain 25–40% fat, especially in PUFAs, in more significant amounts, i.e., 83%. These include omega-3 α-linolenic acid (64%) and omega-6 α-linolenic acid (20%). The ratio of w-3 and w-6 is about 0.32–0.35. As chia seeds possess outstanding fatty acids composition, these are considered a “powerhouse of omega fatty acids”. In contrast, quinoa contains 10-fold lower contents of omega-3 of up to 6.7% of fat than chia seeds [2,27]. In addition, the high amount of polyunsaturated fatty acids helps in improving immunity and preventing heart disease [81]. α-Linolenic acid significantly modulates the expression of lipolytic enzymes, levels of glucose transporter-4, signalling of insulin, and collagen deposition in diet-induced rodents or related diseases [83,84].

Chia seeds are a rich source of vitamins such as Vitamin B1, Vitamin B2, and Niacin in concentrations of 0.6, 0.2, and 8.8 mg/100 g, respectively [25]. The seeds also contain many minerals, out of which potassium, magnesium, calcium, and phosphorus are present in the highest amounts, i.e., 407–726, 335–449, 456–631, 860–919 mg/100 g, respectively [25]. Therefore, it contributes a higher amount of nutrients, i.e., calcium (6 times), potassium (4 times), phosphorus (11 times) than 100 g of milk, and more nutrients, including calcium (13–354 times), potassium (1.6–9.0 times), phosphorus (2.0–12 times) than 100 g of wheat, oats, corn, and rice. Also, chia seeds have 1.8, 6.0, and 2.4 times more iron than lentils, spinach, and liver [82].

### 3.2. Bioactive Compounds in Chia and Its Biological Properties

Chia seeds’ bioactive potential is due to phenolic compounds with antioxidant properties (Figure 5). Antioxidants are the molecules that help scavenge free radicals to prevent oxidative damage in the human body, which can lead to oxidative stress that ultimately causes various chronic diseases such as diabetes, inflammation, and cancer [85,86]. Antioxidants in chia seeds are found in free and bound forms with sugars having glycosidic linkages [87]. Tocopherols, carotenoids, phytosterols, and polyphenolic compounds are the natural antioxidants in chia seeds that promote health benefits against various diseases such as diabetes, Parkinson’s, Alzheimer’s diseases, and different types of cancer [27]. In addition, vitamin E, tannins, and phytates are also present as antioxidant compounds in chia seeds [88]. Children, the elderly, high-performance athletes, women who are prone to osteoporosis, anaemia, patients with diabetes, obesity, or celiac disease are just a few lactose-intolerant consumers who can benefit from quinoa’s qualities, such as its high nutritional value, therapeutic properties, and gluten-free status [29,39,89]. Although there is data on the medicinal potential of quinoa in both animal and human clinical trials, numerous researchers have found several health advantages to quinoa eating. In a nutrition study on children, primarily boys aged 4–5 years from low-income families living in Ecuador, it was discovered that adding quinoa to infant food formulations (about 100 g twice daily for 15 days) significantly increased plasma IGF-1 levels, whereas IGF-1 levels in the control group were unaffected. IGF-1, a hepatic peptide, stimulates growth by increasing bone length and body weight. Malnutrition has been linked to IGF-1 as a marker. Chia seeds do not encourage placebo effects. Seeds do not impact weight loss or disease risk factors in overweight adults with elevated inflammatory markers, blood pressure, or body mass index. Chia-prepared protein and other vital elements in infant feeding are sufficient and help prevent child malnutrition [90]. Natural products specifically altered endogenous antioxidant systems, such as transcription factor Nrf2, via inducing gene transcription associated with antioxidant response elements [91]. Human studies indicated that foods/seeds rich in polyphenols benefit health [92]. Primarily, in vitro studies have widely demonstrated the antioxidant properties of polyphenols; still, the in vivo biological relevance of bioactive polyphenols is questionable especially in cardiovascular due to their low concentration in blood as compared to other antioxidants because of extensive metabolism after ingestion might decline such activity [91,92,93].

#### 3.2.1. Phenolics

Chia seeds (dry basis) possess 8.8% phenolic compounds, including *p*-coumaric acid, caffeic acid, chlorogenic acid, quercetin, rosmarinic acid, cinnamic, gallic, myricetin, and kaempferol, which are present in more significant amounts. The primary phenolics content in chia seeds (in µg per 100 g), includes caffeic acid (27.0), quercetin (0.17), kaempferol (0.01), daidzin (6.6), glycitin (1.4), and genistin (3.4) [27]. In contrast, isoflavones, including glycitein, diadzein, and genistein are in lesser amounts. Flavonoids in chia seeds, such as caffeic acid, chlorogenic acid, and quercetin, provide anti-carcinogenic and anti-hypertensive properties [27]. Chlorogenic acid, caffeic acid, myricetin, kaempferol, and quercetin present in chia seeds offer antioxidant properties, such as caffeic, and chlorogenic acids in preventing peroxidation of fats by scavenging the free radicals and quercetin in protecting against oxidation of fats, proteins, and DNA [81]. Also, rosmarinic acid, found in higher amounts, acts as the most abundant antioxidant present in chia seeds, i.e., 0.93 mg/g, followed by three other phenolic compounds, i.e., protocatechuic acid, caffeic acid, and gallic acid with concentrations of 0. 75, 0.03, and 0.01 mg/g, respectively [87,88,94]. In chia seeds, the phenolic compounds consist of ~98 mg total phenols (GAE/100 g). Antioxidant properties in chia seeds play a vital role in protecting the oil’s bioactive compounds during storage and thermal processing [25]. Also, preserving the bioactive compounds in the chia seeds is essential. Polyphenols, principally cinnamic acid derivatives, and flavonoids are responsible for antioxidant activity in chia seeds. Maintaining the temperature during the processing of chia seeds is crucial as chia seeds heated at a temperature above 90 °C led to a decrease in phenolic compounds such as myrcetin, rosmarinic acid, 3,4-dihydroxybenzoic acid, gallic acid, and caffeic acid [95]. In addition, chia seeds germination is an effective and inexpensive method that can help provide functional chia seed flour by increasing the phenolic compounds, antioxidant activity, and γ-aminobutyric acid (GABA). GABA is a non-protein amino acid that acts as an inhibitory neurotransmitter in the central nervous system and possesses various medicinal properties such as anti-inflammatory, hypocholesterolemic, anti-diabetic, reducing depression, and antiproliferative activity against cancer cells. Digested proteins of chia seeds exhibit anti-inflammatory and anti-adipogenic effects [88].

#### 3.2.2. Phytosterols

Phytosterols are cholesterol-corresponding compounds primarily involved in cholesterol metabolism and helpful in combating cancer, hepatoprotective diabetes, and cardiovascular diseases. Chia seeds are rich in oil ~30–39% and contain phytosterols (~0.02%) such as campesterol, stigmastanol, stigmasterol, Δ5-avenasterol, and β-sitosterol [95]. Stigmasterol is vital to human health due to its prospective anti-osteoarthritic properties and cholesterol-lowering activity that can effectively reduce cardiovascular disease risks. In mechanism, stigmasterol reduces matrix degradation and pro-inflammatory mediators involved in osteoarthritis-induced cartilage degradation via NF-kappa B pathway inhibition [96]. Stigmastanol is engaged in lowering cholesterol absorption from the diet, therefore inhibiting cholesterol biosynthesis within the liver. It shows antidiabetic activity by declining fasting glucose and insulin levels in serum. Stigmastanol has potential anti-parasitic activities against *Leishmania*, *Trypanosoma*, and *Candida* spp. [97]. In addition, it exhibits anticancer properties against various tumorous cells by activating proapoptotic protein signals to suppress the aggregation of cells. β-sitosterol lowers inflammatory activity by inhibiting the STAT1 pathway, and the activation of NF-kB translocation is mediated by tyrosine phosphatase SHP-1. It is beneficial as a dietary supplement to reduce plasma cholesterol and exhibits antioxidant activities or stimulates antioxidant enzymes [98]. In addition, β-sitosterol shows numerous health benefits such as antimicrobial, anti-pulmonary tuberculosis, antidiabetic, anticancer, immune modulation and anti-HIV, anti-arthritic, antipyretic, antihyperlipidemic, anti-atherosclerosis, and angiogenic influences. Chia-seeds-based bioactive exhibit antioxidant activity by scavenging free radicals, chelating ions, or donating hydrogen molecules [68,95].

#### 3.2.3. Polysaccharides

Chia seeds have mucilage consisting of polysaccharides such as 4-O-methyl-α-D-glucopyranosyluronic acid, α-D-glucopyranosyl acid, and β-D-xylopyranosyl acid. It consists of ~48% of total sugars and 4% of protein with high solubility of ~85%, exhibiting high stability up to 240 °C. These compounds, hydrophilic functional groups, are associated with the hydrogel network formation of mucilage [23,99]. It can be used in the food industry as an emulsifier, foam stabilizer, or binder and coating agent for improving the functional properties of foods [100].

#### 3.2.4. Other Bioactives

Chia seeds are depicted as the golden seed of the 21st century [95]. It contains essential unsaturated fatty acids, amino acids, vitamins, and minerals. Chia seeds also provide several health advantages. Linolenic acid, a component of chia seeds, contributes significantly to the formation of leukotrienes and thromboxanes, which are thought to be connected to several significant biochemical processes and physiological functions of the body [85]. Moreover, the omega-3 fatty acid-based molecule can restrict salt and calcium, amplify the parasympathetic nervous system, and resolve channel issues (which might induce hypertension), strengthening ventricular arrhythmia defense [85]. Furthermore, chia seeds during pregnancy promote healthy brain and retinal development in the fetus. Additionally, chia is a strong contender for controlling diabetes since it contains dietary fiber (18–30%), and α-linolenic fatty acids. Chia seeds have a high caloric value of 486 kcal and contain ~2.3-, 8.3-, and 9.8-fold higher fiber per 100 g than oats, corn, and rice. Various reports have widely validated the bioactive supplementation of chia seeds into health benefits such as antioxidant, cardioprotective, anti-inflammatory, neuroprotective, anticoagulant, hypolipemic, hypotensive, hepatoprotective, and hypoglycemic properties [95]. Chia seeds are rich in proteins (15–25%), and bioactive peptides derived after enzymatic hydrolysis can exhibit antioxidant properties. Also, these peptides (16 kDa) show an antihypertensive influence on angiotensin I-converting enzyme (ACE) [27,95]. Albumin and globulin-based peptides primarily exhibit anti-radical potential and reduce DPPH activity. In chia seeds, low bioactive contents such as carotenoids, phytates, and tannins are also present [67].

## 4. Quinoa and Chia Seeds

### 4.1. Processing Procedures

After harvesting quinoa and chia seeds, it becomes essential for further processing to obtain smooth and safe quinoa and chia seeds. Postharvest processing occurs in a series of steps involving the sorting and sifting seeds, sorting based on size, and removal of impurities as well as saponins by utilizing various processing methods. Processing quinoa for contaminants and saponins removal involves several approaches, including polishing, milling, cooking, germination, soaking, boiling, and steaming [101]. Processing is vital in enhancing quinoa’s nutritional properties, promoting consumers’ improved health conditions, and other applications [102,103]. Different processing methods impart other effects on the nutritional profile of quinoa. Raw quinoa flour and germinated or sprouted quinoa flour yield high fiber and iron content as compared to the boiling and roasting of quinoa, which decreases the nutritional content, thus making germinated quinoa flour ideal for inclusion in the diet of anemic and heart patients [104]. Although quinoa possesses various nutrients to prevent the onset of diseases, its availability is retarded due to several anti-nutritional factors or inhibitors such as phytates. Valencia et al. [105] experimented with improving the iron bioavailability of quinoa with the help of processing methods such as soaking, germination, cooking, and fermentation. The results revealed that the fermentation method in germinated quinoa flour is most effective in decreasing the phytate content in quinoa, and thus, enhances iron availability by five to eight times in consumers [106]. Furthermore, quinoa processing is also essential to remove the saponins on the surface of quinoa seeds and reduce the bitterness caused by them [107]. These saponins are present on the surface of grains in a thin layer of glucoside compounds, which is also toxic to health due to their hemolytic activity. Thus, the processing is required to remove the saponins; however, processing methods alter quinoa’s nutritional composition and properties [101]. For eliminating the saponins in quinoa, pearling and sprouting are the most widely utilized methods. However, pearling is the primary method employed nowadays, due to the more straightforward sprouting process of quinoa, which acts as an alternative method to pearling, leading to a decrease in the saponins content thereby reducing bitterness and also enhancing the nutritional as well as sensory characteristics of quinoa and its products such as quinoa-fortified bread [108]. One study was conducted on the effect of different processing methods, including boiling, extrusion cooking, baking, heating under pressure, and dehulling on the quinoa to assess the nutritional quality in terms of essential amino acid content, fatty acid content, and hydrolysis of starch from quinoa. The results revealed that boiling quinoa could be an ideal processing method for retaining essential amino and fatty acids [109]. The thermal processing of quinoa helps improve the quality of quinoa in terms of reduction in phytates, enhanced bioactive constituents, enhanced antioxidative properties, and enhanced hydration properties [110].

Considering the chia seeds, processing methods such as sprouting, soaking, fermentation, and milling help enhance nutrient bioavailability and digestibility. Calvo-Lerma et al. [111] demonstrated the impact of processing methods on the digestion potential of chia seeds. The results conveyed that milling can be ideal for improving chia seeds’ macro and micronutrient potential. In contrast, sprouting improves protein digestibility by decreasing lipolysis in chia seeds. Milling is an effective method as it helps disrupt the intestinal structure of the seeds and leads to enhanced digestibility. Moreover, the sprouting of chia seeds is widely employed as it is a more economical and feasible method that helps to improve the nutritional profile of chia seeds, especially minerals and phenolic content [112]. Furthermore, heating chia seeds or oil in a microwave, such as roasting, also affects its physicochemical properties, including fatty acid, phenolic, and antioxidant activity [27]. Otondi et al. [112] studied the effect of adding chia seed flour with cassava for instant porridge preparation. They found that chai seed blends exhibited high production of instant porridge flour with improved physical and functional properties.

### 4.2. Health Benefits

*S. hispanica* L. and *C. quinoa* L. are plants with distinctive functional and bioactive traits in their seeds and leaves. Because they contain significantly more nutrients and bioactive components than other grains, they are excellent and valuable against physiological disorders like diabetes, hypertension, cardiovascular diseases, and obesity [13,15,26]. For decades, these have been cultivated extensively as a lucrative secondary grain crop for human and animal consumption and a green vegetable. It contains a lot of protein, a balanced range of amino acids, and a lot of lysine (5.1–6.4%), and methionine (0.4–1.0%) contains vitamins (A, C, and E) and many types of minerals [89,113]. *Chenopodium album* is a functional food grain because it is rich in phenolic compounds and flavonoids with high antioxidant activity, glycosides (kaempferol, rutin, and quercetin) [114]. Quinoa is well known for its nutritional properties, due to which it possesses various health benefits such as anti-obesity, hypocholesterolemic effect, antioxidant properties, and cardiovascular diseases [115]. Also, quinoa seeds provide medicinal benefits to higher-risk groups such as children, the elderly, and people with anemia, lactose intolerance, and celiac disease [59,116]. Various studies have been conducted on the beneficial effects of quinoa. One such study was conducted in which quinoa baby food was given to kids for 15 days, resulting in increased plasma-insulin-like growth factor and sufficient protein and other nutrition to prevent malnutrition [29,90]. Also, it has been studied that quinoa helps prevent oxidative stress in animals due to its higher antioxidant capacity by preventing lipid peroxidation in the plasma and tissues of the animals [24].

Chia seeds provide various medicinal health benefits such as anti-hyperlipidemic, anti-diabetic, anti-cancer, anti-inflammatory, and antioxidant properties due to their rich nutritional profile, especially the best plant source of omega-3 fatty acids (α-linolenic acid) [117,118,119]. Omega-3 fatty acids are very useful in preventing hypertension by blocking calcium and sodium channel dysfunctions, which can prevent ventricular arrhythmia that causes the heart to beat very fast to avoid the transportation of oxygen-rich blood to the brain and body that leads to cardiac arrest. Also, the consumption of chia seeds during pregnancy leads to the development of the brain and retina of the fetus [27]. In addition, chia seeds are also a rich source of dietary fiber, providing various medicinal benefits as fiber in the diet increases stool volume and prevents diseases such as diverticulosis and even cancer. The fiber content in chia seeds also plays a vital role in treating diabetes by slowing down the release of glucose and the digestion process in the body [120]. Aguirre et al. [121] studied adding chia seeds, quinoa flour, and wheat flour for bread making. They showed better sensory quality attributes and nutritional properties regarding proteins, fats, and carbohydrates. Bioactive peptides from chia seeds have also been separated, isolated, identified, and proven to exert health benefits such as antioxidant and anti-inflammatory activity, antihypertensive, antidiabetic, and antimicrobial [122,123].

### 4.3. Value-Added Food Products and Other Potential Applications

Due to its nutritional benefits, quinoa is widely used in supplements such as soups, cakes, biscuits, breakfast cereal, alcohol, and baby foods [124,125,126,127]. In addition, it can be consumed as an alternative to rice in sprouts and popcorn. Quinoa-flour-incorporated noodles provide a different and unique alternative to celiac disease patients compared to usual noodles. Also, quinoa flakes are widely considered a new product that can be prepared by drying and pressing between converging rollers, and these flakes are further used in the preparation of soups, cakes, juices, and pies. Extruded quinoa products are also beneficial as the extrusion process retains the nutritional value of the quinoa due to less processing time than puffed quinoa. The extrusion process utilizes high temperature and pressure for a short time by causing starch gelatinization, dextrinization, and protein restructuring to give a different texture to the final product [26].

Stikic et al. conducted a study to develop bread by incorporating quinoa seeds. The result showed increased nutritional content, i.e., 2% higher protein and 1% higher oil and fiber content, along with excellent sensory characteristics in the fortified bread [128]. Li et al. reported that incorporating 20% of quinoa flour in bread manufacture could deliver health benefits to consumers [129]. Similarly, Srujana et al. demonstrated findings on developing traditional gluten-free recipes of Indian ladoo, and chapattis by substituting germinated quinoa flour in different proportions, i.e., 25, 50, 75, and 100% [130]. The result was evaluated regarding sensory attributes and indicated that 25% of germinated quinoa flour leads to acceptable ladoo and chapattis. Quinoa in flour can also be used to prepare other nutritious baked products such as biscuits. Fortifying quinoa flour with wheat flour helps increase the biscuits’ protein, fiber, and ash content to provide highly healthy biscuits to consumers [131]. Wheat bread is nutritionally enhanced by quinoa flour supplemented dose-dependently [132]. Cranberry jam’s antioxidant properties are improved by the co-supplementation of chia seeds and gold flax [133].

According to USA dietary guidelines in 2000, chia seeds can be used as a substitute in food products but in smaller quantities, i.e., not more than 48 g/day. Chia seeds are incorporated in various food products such as pasta, biscuits, cereals, cakes, and snacks [134,135]. Chia seeds/gels can be used as an alternative to eggs and fat due to their hydrophilic property. Also, chia oil can be substituted for 25% of egg quantity in cakes [27]. Chia gel is a substitute that helps decrease food products’ calorie and fat content. In addition, adding chia seeds to baked products helps increase omega-3 fatty acids essential for healthy living [25]. A study was conducted by Sayed-Ahmad et al. on developing whole wheat bread by incorporating chia flour and analyzing its nutritional and technological characteristics [136]. The results claim that the bread becomes softer with good sensory characteristics, increased antioxidant activity, and nutritional value after adding chia flour. The supplementation of chia oil improves oxidative stress of the liver via transcription factors PPAR-γ and Nrf2 upregulation involving antioxidant responses [137]. In addition, chia seeds ameliorate endothelial dysfunction in metabolic syndrome, liver inflammation or injury, and oxidative stress [138,139]. Extracted mucilage from chia seed showed a prominence of novel biological activities, such as mucoadhesive properties that can be potentially applied for developing various pharmaceutical and cosmetic products [140].

Another study was conducted by Oliveira et al. on the development of pasta for replacing wheat flour with chia flour to analyze its nutritional and technological properties [141]. The results revealed that 7.5% of chia flour incorporated in pasta leads to increased nutritive value with higher sensory acceptability. Puri et al. demonstrated the assessment of the nutritional value of snacks (matthi) prepared by fortification with gram flour and chia seeds by replacing wheat flour [131]. Four proportions were made for gram flour and chia seed flour, i.e., 50:0, 35:15, 25:25, and 15:35, respectively. The results showed an increase in fiber and mineral content of matthi after adding chia seed flour and thus indicate a broad scope for developing snacks from chia seeds. Chia seeds are widely used in cookies, bread, and sweets due to their high fiber content which contributes to increased water holding, absorption, and emulsion capacity. Also due to their high protein, low carbohydrate, and high insoluble dietary fiber content, chia seeds are fortified in bakery products for diabetic and obese patients [142]. Chia seeds can also be fortified in jams such as pineapple jam which can provide a taste with high nutritional value, i.e., increased protein and fiber content [143]. Punia and Dhull have supplemented cookies with 20% chia seed mucilage; it possessed the best texture and was found to be acceptable regarding colour and mouthfeel [144]. Yoghurt with 6% chia seeds and 12% strawberry showed increased levels of protein, lipids, fibre, PUFA, especially omega 3 fatty acids and minerals with acceptable sensory quality and probiotic properties [145]. In addition to the studies of various food products by incorporating quinoa and chia seeds separately, multiple studies have been performed with the combination of quinoa and chia seeds. Goyat et al. demonstrated the development and evaluation of nutritional, functional, and sensory characteristics of ready-to-eat porridge by fortification with chia and quinoa seeds in a 1:1 ratio by replacing rice flour in three different proportions, i.e., 10, 15, and 20% [146]. The results revealed that a 15% replacement of rice flour with quinoa and chia seeds in a 1:1 ratio was more acceptable regarding sensory and nutritional value. Approaches to enhance the food composition through avoiding risks associated with transgenic crops to their low acceptance by communities are under broad consideration [147]. Antioxidant vitamin compositions of dry, germinating seeds and sprout of chia examination suggested that dry seeds contain eight-fold higher vitamin E over sprouts. After imbibition, the seed’s vitamin C contents increased to 17.5-fold more than sprouts [148]. Chia oil improves the lipid profiles of beef-based burgers. It can replace the use of animal fat, and mucilage applied as a versatile function food additive, or food ingredient replacer [148,149,150]. Chia seed contains trypsin inhibitors that can be potentially employed as antimicrobial agents against methicillin-resistant *Staphylococcus aureus* [151]. Antioxidant properties of yoghurts are highly facilitated by chia seeds and after soaked in apple juice [152]. Chia-seeds-based natural gum can be used in vegan mayonnaise [153]. In addition, the extract of chia seeds is employed to synthesize biogenic nanomaterials for therapeutic applications such as anticancer activity, cytotoxicity, and photocatalytic activity [154,155]. The flavonoid and phenolic contents of chia sprouts are significantly increased up to 6.4- and 11.5-fold on germination with dry seeds, respectively [156]. Under similar conditions, the sprout’s antioxidant activity enhanced up to 29-fold. The sprouts bioactivity showed potent antimicrobial activity towards enteric bacterial pathogens such as *Escherichia coli*, *Pseudomonas aeruginosa*, *Salmonella typhi*, and *S. aureus* [156].

## 5. Conclusions

Quinoa and chia seeds are essential superfoods for health and wellness. Quinoa is rich in protein and its quality is like casein in milk, as it consists of all nine essential amino acids required for healthy living. Also, quinoa possesses antioxidant and anticancer activities due to the presence of various bioactive compounds such as kaempferol. There are many different food products in which quinoa is utilized as a substitute to enhance the nutritional property of the food products, including extruded snacks, traditional foods, meat products, and bakery products. In addition, chia seeds also play a vital role in human health because of their high omega-3 fatty acid content and phytochemicals which provide antioxidant action and are essential in preventing various chronic diseases such as heart disease. Therefore, chia seeds are substituted in multiple forms in food products to increase their nutritional value. Furthermore, both quinoa and chia seeds are gluten-free which makes them ideal for consumption by celiac disease patients. Additionally, many challenges can occur when including quinoa and chia seeds in food products, which need to be studied. So, there is a need for further studies to reveal its benefits to a larger population and allow researchers to investigate more of their effects on human health.

## Figures and Tables

**Figure 1 antioxidants-12-01413-f001:**
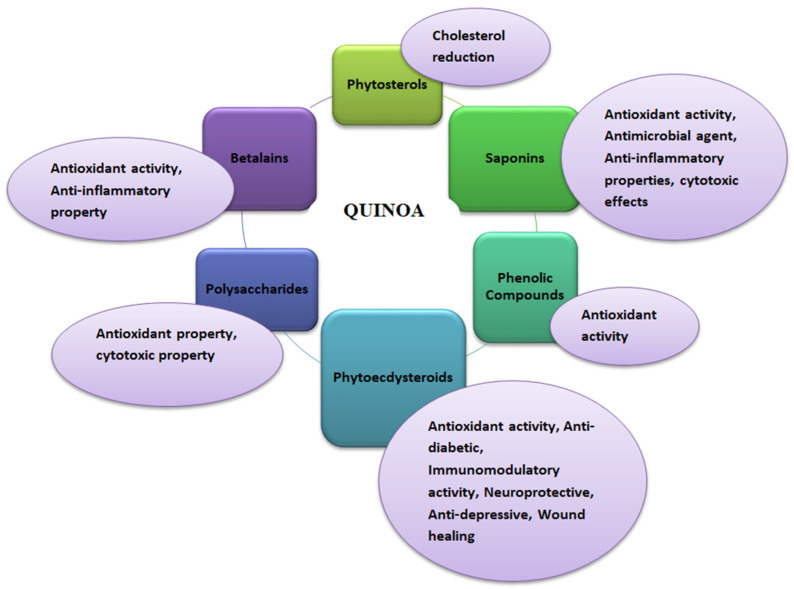
Functional properties of quinoa (*Chenopodium quinoa* Willd).

**Figure 2 antioxidants-12-01413-f002:**
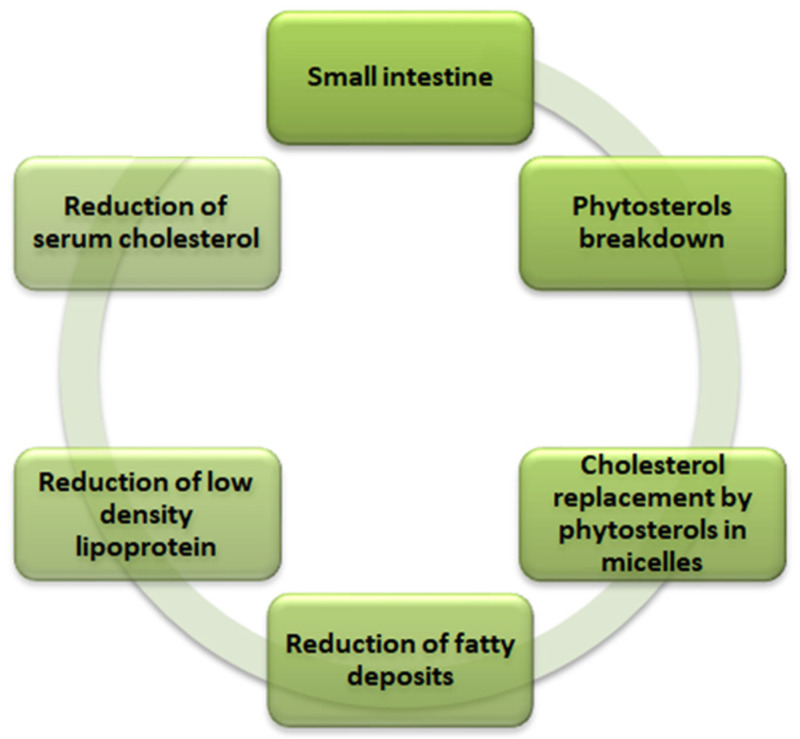
Mechanism of cholesterol reduction in human body by phytosterols in quinoa (*Chenopodium quinoa* Willd).

**Figure 3 antioxidants-12-01413-f003:**
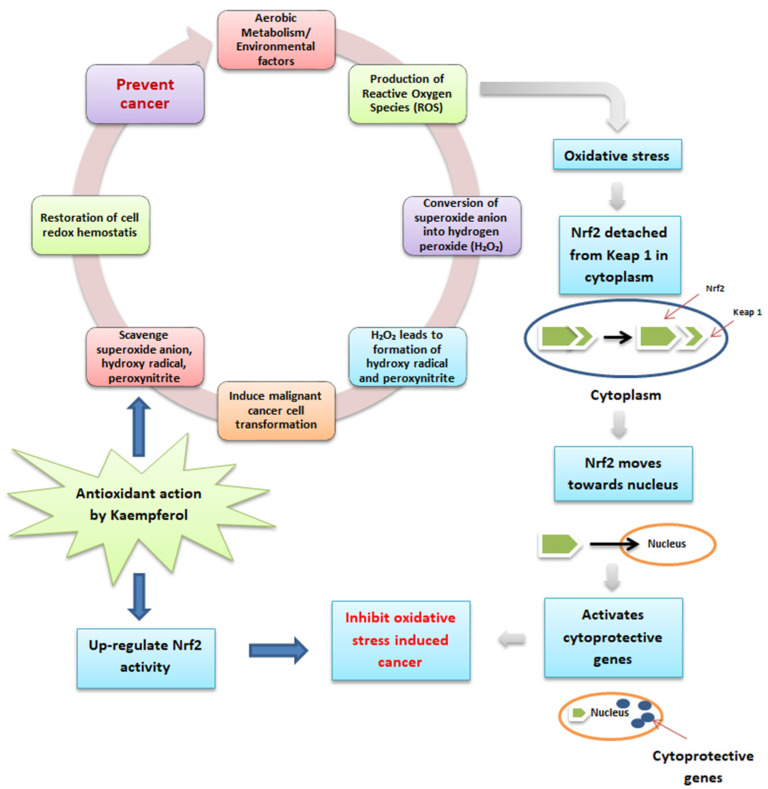
Antioxidant activity and mode of action of kaempferol in preventing cancer.

**Figure 4 antioxidants-12-01413-f004:**
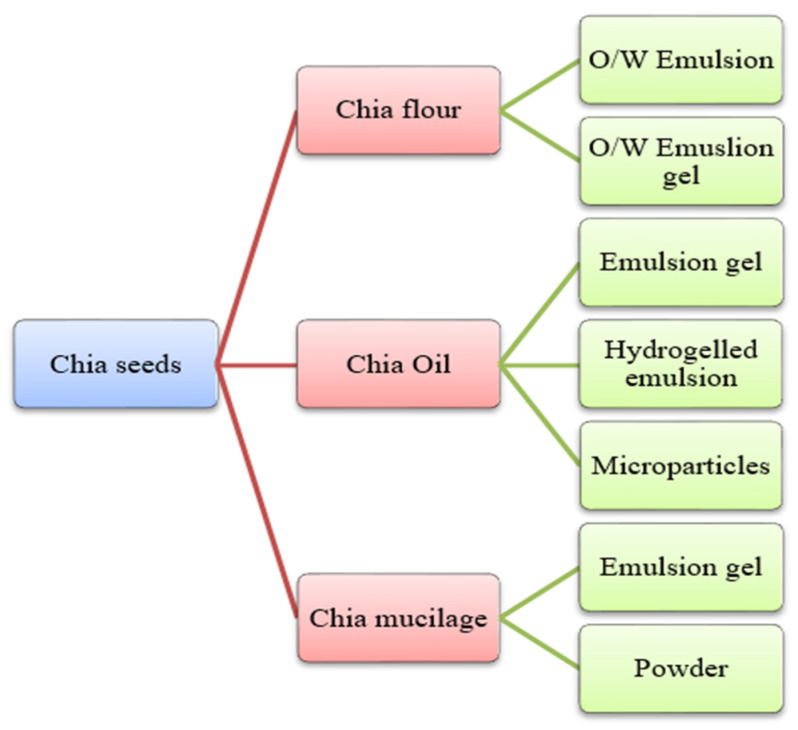
Different forms of chia (*Salvia hispanica*) seeds are used in food products.

**Figure 5 antioxidants-12-01413-f005:**
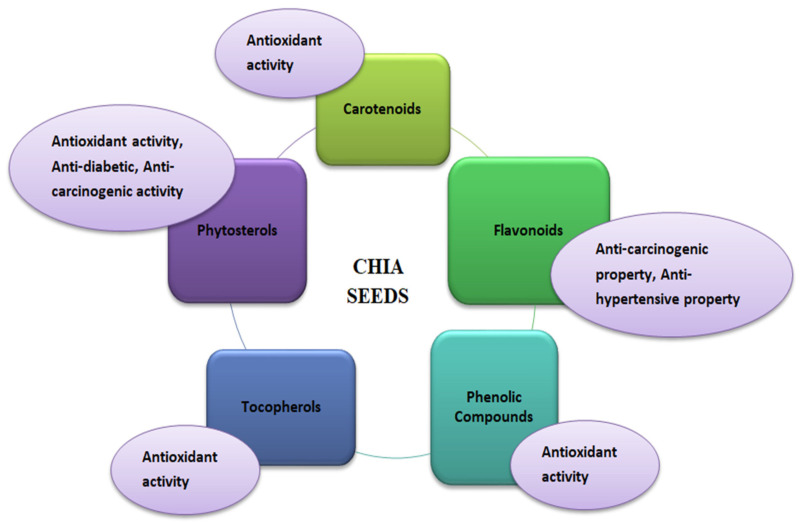
Functional properties of chia (*Salvia hispanica*) seeds.

**Table 2 antioxidants-12-01413-t002:** Bioactive compounds and antioxidant activity of quinoa and chia seeds.

Bioactive Compounds	Unit	Availability (Per 100 g)		Reference
		Quinoa	Chia	
Phytosterols	mg	38.0–118	153–497	[24,67,68]
Saponins	mg	20–3400	+ ^a^	[24,26]
Phenolics	mg	46.0–202	3.5–253	[27,67]
Phyotoecdysteroids	mg	13.8–57.0	+	[24]
Betalains	µg	150–610	+	[24]
Flavonoids	mg	25.9–289	62.5–122	[28,51]
Carotenoids	µg	0.65–1.81	13.4–39.8	[57,67]
Squalene	µg	148–256	111–299	[58,67]
Tocopherols	mg	0.4–52.0	35.6–60.0	[51,57,67,68]
Oil	%	2.00-10-0	28.5–32.7	[9,10]
Linolenic acid (omega 3)	%	6.5–6.7	59.8–63.8	[27,58]
Linoleic acid (omega 6)	%	56.4–60.1	18.9–20.4	[27,58]
Antioxidant activity (DPPH)	mg TE/g	1.3–6.0	1.60–109	[23]
Antioxidant activity (FRAP)	mg TE/g	0.7–9.0	5.10–278	[23]

^a^ trace amounts.

**Table 3 antioxidants-12-01413-t003:** Extraction procedure for quinoa and chia oils [16,23,27,73].

Extraction Method	Solvent
Cold solvent	*n*-hexane
Soxhlet	Ethyl acetate ethanol/*n*-hexane
Ultrasonic/Soxhlet	*n*-hexane/ethyl acetate/isopropanol
Ultrasound	Ethyl acetate/ethanol
Cold pressing	Ethanol
Ultrasound liquid-liquid	Methanol-water solution
Hot solvent	Water and aqueous ethanol
Ultrasound-assisted	*n*-hexane
Cold pressing and ultrasound	Methanol
Screw pressing	*n*-hexane
Supercritical fluid	Carbon dioxide/ethanol
Pressurized liquid	Ethanol/*n*-hexane
Subcritical fluid	*n*-propane

## Data Availability

Not applicable.

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
