# Peer review of "Nutritional and Functional New Perspectives and Potential Health Benefits of Quinoa and Chia Seeds"

_antioxidants, 2023, doi:10.3390/antiox12071413_

Round 1
Reviewer 1 Report (Previous Reviewer 1)
This paper is suitable for pubilication.
Author Response
Title: Nutritional and Functional New Perspectives and Potential Health Benefits of Quinoa and Chia Seeds
Reviewer #1:
Comments
This paper is suitable for pubilication.
Response: The authors would like to thank reviewer#1 for considering our work.

Reviewer 2 Report (Previous Reviewer 2)
Review:
“Nutritional and Functional New Perspectives and Potential Health Benefits of Quinoa and Chia Seeds “
I was a reviewer of the first draft of this work and have to say it has improved a lot. Table 1,2 and 3 are achievements that make comparing both crops now very inviting. In Table 2 we have a column of references. Just as a suggestion, coherence could be improved by adding such a column also to table 3.
The figures also make the article attractive and one has now the feeling to read it with gain.
However, the use of the English language is still “creative” at times. The authors have to pay attention to really say what they mean. In this regard, they still have to go through the text with a fine comb.
Just as one single example: Line 638:
“However, their awareness of health's functional and medicinal properties is not known globally.”
This sentence makes no sense (although it’s clear what the authors meant to say.) I really hope that this aspect of the article will also improve.
However, the use of the English language is still “creative” at times. The authors have to pay attention to really say what they mean. In this regard, they still have to go through the text with a fine comb.
Just as one single example: Line 638:
“However, their awareness of health's functional and medicinal properties is not known globally.”
This sentence makes no sense (although it’s clear what the authors meant to say.) I really hope that this aspect of the article will also improve.
Author Response
Manuscript: antioxidants-2488518
Title: Nutritional and Functional New Perspectives and Potential Health Benefits of Quinoa and Chia Seeds
Reviewer #2:
“Nutritional and Functional New Perspectives and Potential Health Benefits of Quinoa and Chia Seeds “
I was a reviewer of the first draft of this work and have to say it has improved a lot. Table 1,2 and 3 are achievements that make comparing both crops now very inviting. In Table 2 we have a column of references. Just as a suggestion, coherence could be improved by adding such a column also to table 3.
Response: Authors would like to thank reviewer#2 for his/her valuable feedback on the manuscript. Per the reviewer’s suggestions, the reference has been provided as follows:
(line, 290) “Table 3. Extraction procedure for quinoa and chia oils [16,23,27,73]
The figures also make the article attractive and one has now the feeling to read it with gain.
However, the use of the English language is still “creative” at times. The authors have to pay attention to really say what they mean. In this regard, they still have to go through the text with a fine comb.
Just as one single example: Line 638:
“However, their awareness of health's functional and medicinal properties is not known globally.”
This sentence makes no sense (although it’s clear what the authors meant to say.) I really hope that this aspect of the article will also improve.
Response: As per the reviewer’s suggestion, the information has been deleted (line, 638).

Reviewer 3 Report (Previous Reviewer 3)
Authors reported to review herein many bioactive compounds in the seeds both quinoa and chia have human health benefits to help dietary and treat human diseases including cancers. This resubmits article can help to research for developing and understanding natural products, especially focused on quinoa and chia. This report showed composition and health benefits of the nutrients in both materials. It is recommended to publish to the Journal with minor corrections as below.
From line 20 to Line 26 as below: needed to rewrite with appropriate words and sentences.
Line 20, In contrast,
Line 22, In addition,
Line 26, In addition,
Line 538, change “AAplications” to Applications
Quinoa is known for its high and good quality protein content and nine essential amino acids vital for an individual's development and growth. In contrast, chia seeds contain high dietary fiber content, calories, lipids, minerals (calcium, magnesium, iron, phosphorus, and zinc), and vitamins (A and B complex). In addition, chia seeds are known for their presence of a high amount of omega-3 fatty acids. Both quinoa and chia seeds are gluten-free and provide medicinal properties due to bioactive compounds, which help combat various chronic diseases such as diabetes, obesity, cardiovascular diseases, and metabolic diseases such as cancer. In addition, quinoa seeds possess phenolic compounds, particularly kaempferol, that can help prevent cancer.
From line 20 to Line 26 as below: needed to rewrite with appropriate words and sentences.
Line 20, In contrast,
Line 22, In addition,
Line 26, In addition,
Line 538, change “AAplications” to Applications
Quinoa is known for its high and good quality protein content and nine essential amino acids vital for an individual's development and growth. In contrast, chia seeds contain high dietary fiber content, calories, lipids, minerals (calcium, magnesium, iron, phosphorus, and zinc), and vitamins (A and B complex). In addition, chia seeds are known for their presence of a high amount of omega-3 fatty acids. Both quinoa and chia seeds are gluten-free and provide medicinal properties due to bioactive compounds, which help combat various chronic diseases such as diabetes, obesity, cardiovascular diseases, and metabolic diseases such as cancer. In addition, quinoa seeds possess phenolic compounds, particularly kaempferol, that can help prevent cancer.
Author Response
Manuscript: antioxidants-2488518
Title: Nutritional and Functional New Perspectives and Potential Health Benefits of Quinoa and Chia Seeds
Reviewer #3:
Authors reported to review herein many bioactive compounds in the seeds both quinoa and chia have human health benefits to help dietary and treat human diseases including cancers. This resubmits article can help to research for developing and understanding natural products, especially focused on quinoa and chia. This report showed composition and health benefits of the nutrients in both materials. It is recommended to publish to the Journal with minor corrections as below.
Response: Authors would like to appreciate reviewer#3 for his/her valuable feedback on the manuscript.
From line 20 to Line 26 as below: needed to rewrite with appropriate words and sentences.
Line 20, In contrast,
Line 22, In addition,
Line 26, In addition,
Quinoa is known for its high and good quality protein content and nine essential amino acids vital for an individual's development and growth. In contrast, chia seeds contain high dietary fiber content, calories, lipids, minerals (calcium, magnesium, iron, phosphorus, and zinc), and vitamins (A and B complex). In addition, chia seeds are known for their presence of a high amount of omega-3 fatty acids. Both quinoa and chia seeds are gluten-free and provide medicinal properties due to bioactive compounds, which help combat various chronic diseases such as diabetes, obesity, cardiovascular diseases, and metabolic diseases such as cancer. In addition, quinoa seeds possess phenolic compounds, particularly kaempferol, that can help prevent cancer.
Response: As per the reviewer’s suggestion, the information has been revised as follows:
(line, 20) “Whereas, ~.
(line, 22) “Chia seeds are also known ~.
(lines, 25-26) “Quinoa seeds possess ~.
Line 538, change “AAplications” to Applications
Response: The information has been rectified (line, 538).

This manuscript is a resubmission of an earlier submission. The following is a list of the peer review reports and author responses from that submission.
Round 1
Reviewer 1 Report
Major points
The paper aimed to review health beneficial values of quinona and chia seeds in various aspects including nutrients, bioactive ingredients, food processing, and value-added food production. It contains lots of information sufficient to make the world aware of the usefulness of their seeds. However, information on antioxidants seems rather poor and superficial, although there are some antioxidative ingredients such as quercetin, kaempferol, and other phenolic antioxidants described in the context. In in vitro, it is true that phenolic antioxidants act as direct scavenger of ROS via the auto-oxidation of phenolic hydroxyl group. However, in vivo, most scientists now believe that phenolic compounds likely act as indirect antioxidants, e.g. via Nrf2 pathway as the authors mentioned a little about it. Please check the following three literatures discussing the polyphenol/antioxidant controversy. So, I strongly recommend the authors to submit the paper to other MDPI journals such as “Foods”, “Nutrients” and “Nutraceuticals”, which apparently seem more suitable to the paper.
https://www.sciencedirect.com/science/article/abs/pii/S0003986115003689
https://academic.oup.com/jn/article/141/5/989S/4689148
https://www.ncbi.nlm.nih.gov/pmc/articles/PMC9967135/
Minor points
1) Line 329, 4.2. Bioactive Compounds in Chia Seeds
In the latter part of this section (line 378〜), some effects of quinoa but not chia are mainly discussed.
2) Figure 2
“Reduction of serum cholesterol” looks connected with “Small intestine” through the circle arrow.

Reviewer 2 Report
Review
Title:
I am not convinced of the title:
“Recent Updates on Nutritional and Functional Perspectives of Quinoa and Chia Seeds with Potential Health Benefits.”
By definition it is a review and thus should anyway give an update on new developments in the field, no need to mention that in the title.
“with potential health benefits”. Here we have ambiguity as it sounds like there are also seeds without potential health benefits. A title like: “Nutritional and functional perspectives and potential health benefits of Quinoa and Chia seeds” would probably be more appropriate, but the authors should look for something better than this. One could also include “New” as in “New… perspectives”. In that case we would expect a review of recent developments, especially from targeted research work. The references are indeed mainly from the years 2015-2022 but a distinction between original article and review is not made. I would like to find the phrase “(for a review see ref.xx)” more often in the text.
Author list:
The big list of authors is a little questionable. Reviews are often written by only two authors, here we have six. The family name of the second author seems to be missing; according to the e-mail it should be “Haleem”.
Abstract:
I don’t understand the use of the phrase “on the other hand”.
Do the authors want to tell the reader what distinguishes Chia and Quinoa?
Then we would expect something like “In contrast to Quinoa, Chia seeds contain..
But Quinoa contains “high and good quality protein content” and Chia seeds also.
Thus, the use of the English language seems to be a bit shaky in this article. It is often clear what the authors mean but how they express it is imprecise to say the least. Additional competent editing is required throughout the article, not only in regard to the language but also in regard to the logic of what is said.
(Nowadays we also have nearly perfect online-translation software, for instance “DeepL” I understand that Hindi-English translation might have complexities I am not aware of. But even here websites exist and could give some assistance.
https://www.typingbaba.com/translator/hindi-to-english-translation.php)
Structuring:
The authors review Chia and Quinoa. Why especially these two? What’s the reason for this selection? But because of this selection, the reader now expects a careful comparison of both, highlighting the similarities and differences.
The authors have chosen the following structure of their review.
1. Intro
2. Quinoa Characteristics
2.1 Quinoa Nutrients
3. Quinoa Bioactive Compounds
3.1 Phytosterols
3.2 Saponins
3.3 Phenolics
3.4 Phytoecdysteroids
3.5 Polysaccarides
3.6 Betalains
4. Chia Characteristics
4.1 Chia Nutrients
4.2 Chia Bioactive Compounds
4.3 Processing of Quinoa and Chia
4.4 Health Benefits of Quinoa and Chia
4.5 Value added food products from Quinoa and Chia
5. Conclusions
Thus, in the first part we deal only with Quinoa, in the second part only with Chia and in the third part with both of them together.
It looks like the review was written by two different authors or teams of authors, one dealing with Quinoa, the other with Chia. This becomes especially obvious when one looks at the tables. Table 2 on page 11 follows Table 3 on page 9.
What we would expect from the structure is that all things concerning only Quinoa are structured by 2, 2.1, 2.2., and that all things concerning Chia are structured by 3. 3.1, 3.2, and that all things concerning both of them by 4, 4.1, 4.2 and 4.3.
This lack of parallelism becomes also obvious when we look at the tables.
We have Table 1 listing the nutritional components of Quinoa.
The Chia part starts with Table 3, the extraction of Chia oil. It is not mentioned why this table has no counterpart in the Quinoa section. The counterpart to Table 1 is Table 4 (mislabeled by 2) but the structuring of this table is such that comparison of both “Nutrition tables” is very difficult; whereas the first tables distinguishes explicitly between vitamins and minerals, the second does not.
On page 10, we read about Chia “the richest source of omega-3- fatty acids (72). “ Yet, these fats are not even mentioned in the Chia Seed Table; neither is the water content.
It would be so nice to understand why one may should consume both foods, to understand the ways they complement each other. But presented in this way, this work is mainly left to the reader.
The Quinoa part details the bioactive compounds much more than the Chia part as is instantly obvious from the structure. This also hinders the reader a bit to compare both foods.
I stop here.
I think it has become clear that this review still needs some or a lot more work, and I encourage the authors to put that in.

Reviewer 3 Report
Authors reported herein the seeds of quinoa and chia have some bioactive compounds to help human being to fight human diseases including cancers. Quinoa has well-known having their benefits, but chia doesn’t do too much. This report showed both together in this manuscript. It may be more understandable if contains their chemical structures, as kaempferol and some phenolic compounds. It is recommended to publish to the Journal with minor corrections as below.
Line 22, Change “In addition, Chia” to “In addition, chia”
Line 581, Change “Therefore, Chia” to “Therefore, chia”
Line 101, Change “(>15mm)” to “(> 15 mm)”
Line 102, Change “(5-15mm), and small (<5mm)” to “(5-15 mm), and small (< 5 mm)”
Line 111, Change “[39,40].” to “[39,40]” - no period
Line 285, Change “[63,64].” to “[63,64]” - no period
Line 326, Change “[39,58].” to “[39,58]” - no period
Line 314, change “a-linolenic” to “a-Linolenic”